# Phenotypic and genotypic drug susceptibility patterns of *Mycobacterium tuberculosis* isolates from pulmonary tuberculosis patients in Central and Southern Ethiopia

**Melaku Tilahun**[1,2]*, **Teklu Wegayehu**[1☯], **Biniam Wondale**[1☯], **Tewdros Tariku Gebresilase**[2☯], **Tesfaye Gebreyohannes**[2], **Abraham Tekola**[2], **Mekdes Alemu**[2], **Sebsib Neway**[2], **Bethlehem Adnew**[2], **Maeruf Fetu Nassir**[2], **Yonas Kassahun**[2☯], **Abraham Aseffa**[2☯], **Kidist Bobosha**[2☯]

**1** Department of Biology, College of Natural and Computational Sciences, Arba Minch University, Arba Minch, Ethiopia, **2** Armauer Hansen Research Institute (AHRI), Addis Ababa, Ethiopia

☯ These authors contributed equally to this work.
* mtilahun600@gmail.com

## Abstract

### Introduction

The persistence of tuberculosis (TB) infection in some patients after treatment has highlighted the importance of drug susceptibility testing (DST). This study aimed to determine the drug susceptibility patterns of *Mycobacterium tuberculosis* (*M. tuberculosis*) isolates from pulmonary TB (PTB) patients in Central and Southern Ethiopia.

### Methods

A health institution-based cross-sectional study was conducted between July 2021 and April 2022. Sputum samples were collected from newly diagnosed smear microscopy and/or Xpert MTB/RIF-positive PTB patients. The samples were processed and cultivated in Lowenstein-Jensen (LJ) pyruvate and glycerol medium. *M. tuberculosis* isolates were identified using polymerase chain reaction (PCR) based region of difference 9 (RD9) deletion typing. Phenotypic DST patterns of the isolates were characterized using the BACTEC MGIT™ 960 instrument with SIRE kit. Isoniazid (INH) and Rifampicin (RIF) resistant *M. tuberculosis* isolates were identified using the GenoType® MTBDR*plus* assay.

### Results

Sputum samples were collected from 350 PTB patients, 315 (90%) of which were culture-positive, and phenotypic and genotypic DST were determined for 266 and 261 isolates, respectively. Due to invalid results and missing data, 6% (16/266) of the isolates were excluded, while 94% (250/266) were included in the paired analysis. According to the findings, 14.4% (36/250) of the isolates tested positive for resistance to at least one anti-TB drug. Gene mutations were observed only in the *rpoB* and *katG* gene loci, indicating RIF and high-level INH resistance. The GenoType® MTBDR*plus* assay has a sensitivity of 42%

**Data Availability Statement:** All relevant data are within the paper and its Supporting information files.

**Funding:** This work is supported by Human Heredity and Health in Africa (H3Africa) [Grant number: H3A-18-003] (https://h3africa.org/). KB obtained the study grants. H3Africa is managed by the Science for Africa (SFA) Foundation in partnership with Welcome Trust (UK), National Institute of Health (US), and African Society of Human Genetics (AfSHG). The views expressed herein are those of the author(s) and not necessarily those of the SFA Foundation and her partnership. Part of the work also supported by AHRI Core budget (www.ahri.gov.et). AHRI receives core support from Sida, NORAD and the Government of Ethiopia. There was no additional external funding received for this study. All funders had no role in study design, data collection and analysis, decision to publish, or preparation of the manuscript.

**Competing interests:** The authors have declared that no competing interests exist.

and a specificity of 100% in detecting INH-resistant *M. tuberculosis* isolates, with a kappa value of 0.56 (95%CI: 0.36–0.76) compared to the BACTEC MGIT™ DST. The overall discordance between the two methods was 5.6% (14/250) for INH alone and 0% for RIF resistance and MDR-TB (resistance to both INH and RIF) detection.

## Conclusion

This study reveals a higher prevalence of phenotypic and genotypic discordant INH-resistant *M. tuberculosis* isolates in the study area. The use of whole-genome sequencing (WGS) is essential for gaining a comprehensive understanding of these discrepancies within INH-resistant *M. tuberculosis* strains.

## Introduction

Tuberculosis (TB) is an infectious disease caused by *M. tuberculosis*. It is the main cause of death among infectious diseases until the coronavirus (Covid-19) outbreak [1]. Evidence in the past two decades showed that TB patients, despite taking the full course of the anti-TB drug regimen, remain smear-positive [2]. This may be due to the emergence and transmission of rifampicin resistance (RR) and/or multidrug-resistant (MDR) strains due to poor management of TB cases [3]. In 2020, 71% of cases of pulmonary tuberculosis (PTB) with bacteriological confirmation were tested for RR globally. Among RR tested cases, 132,222 were RR/MDR-TB and 25,681 cases were pre-XDR-TB or XDR-TB (extensive drug resistance) [1]. Ethiopia is one of the 30 nations with a high TB burden with an estimated TB incidence rate of 140/per 100,000 people annually. In 2019, it was anticipated that 7.5% of previously treated TB patients and 1.1% of newly diagnosed TB cases had MDR-TB [4]. Given this, RR/MDR-TB is a public health concern that needs special attention worldwide. MDR-TB is characterized by *M. tuberculosis* resistance to at least two key first-line anti-TB drugs INH and RIF [3]. Resistance to INH is primarily caused by mutations in the *katG* gene, which encodes catalase-peroxidase, that activates INH. Mutations in the *inhA* promoter region result in the upregulation of the drug target InhA, a protein reductase that is important in the biosynthesis of mycolic acid [5]. Resistance to RIF is mainly caused by mutations in the β subunit of the RNA polymerase, which is encoded by the *rpoB* gene [6]. The situation is currently deteriorating as a result of the emergence of XDR-TB strains, which are MDR-TB strains resistant to fluoroquinolones (FQs) and at least one second-line injectable drug (SLID) such as kanamycin, amikacin, and capreomycin [7]. The DNA gyrase, encoded by *gyrA* and *gyrB*, is the main target of FQs in *M. tuberculosis*, and resistance is primarily caused by mutations in two short regions known as "quinolone resistance-determining regions" (QRDRs) genes [8]. Resistance to SLIDs in *M. tuberculosis* has been linked to mutations in the *rr* gene, which encodes the 16S rRNA component of the 30S small subunit of the bacterial ribosome [9]. The identification of drug-resistant strains relies on validating the isolates' drug susceptibility pattern using phenotypic techniques or locating altered drug target genes using genotypic technologies [10]. The use of genotypic techniques for the rapid screening of patients at risk for MDR-TB was supported by the WHO based on evidence and professional opinion [11]. The technique detects *M. tuberculosis* and treatment resistance-conferring gene mutations at the same time, promising to transform patient care and stop transmission by ensuring early diagnosis. However, the presence of phenotypic-genotypic discrepant drug-resistant bacterial populations has decreased the sensitivity of the molecular methods alone [12]. Evidence suggested that the low sensitivity of the

methods limits the global frontline molecular tools, such as GenoType® MTBDR*plus* and Xpert MTB/RIF assay, to detect all MDR-TB cases [13]. Based on the evidence, worldwide, 95% of RIF-resistant mutations are present in the *rpoB* gene [14]. Nevertheless, discordant findings have been reported elsewhere; according to one study, only 61.5% of strains contained mutations in the rifampicin resistance determining region (RRDR) of the *rpoB* gene [15]. This discovery is consistent with the observation that up to 30% of MDR-TB strains have RIF resistance-related mutations outside of RRDR [16]. This implies that there is a possibility of the absence of mutation in RRDR of the *rpoB gene* in MDR-TB isolates due to the existence of other rare *rpoB* mutations outside the region or a different mechanism of RIF resistance. Similarly, inconsistent results between phenotypic and genotypic methods in the diagnosis of INH-resistant *M. tuberculosis* isolates have been reported elsewhere. *Wondale et al.* evaluated the diagnostic performance of the GenoType® MTBDR*plus* assay using BACTEC MGIT™ 960 as a gold standard and discovered that the method's sensitivity to detect INH resistance was 33.3% [17]. Likewise, *Ahmad et al.* support the findings of phenotypic and genotypic method discordance in a low-TB incidence country [18]. This piece of evidence anticipated the existence of rare mutations that were not incorporated in the GenoType® MTBDR*plus* assay strips or by mutations in other genomic loci of the *KatG* and *inhA* genes or by other mechanisms that demand further investigation. Studying TB and DR-TB across vast geographic areas using diverse DST methods provides a valuable opportunity to collect comprehensive data, analyze trends and patterns, and gain a holistic understanding of the scope of the problem. Thus, such a thorough investigation provides valuable insights and strategies for addressing this critical public health issue. Therefore, this particular study aimed to detect the phenotypic and genotypic drug susceptibility patterns of *M. tuberculosis* isolates from PTB patients in Central and Southern Ethiopia.

## Material and methods

### Study design

A health institution-based cross-sectional survey was conducted from July 2021 to April 2022 in the Central and Southern parts of Ethiopia. The study sites were chosen from Oromia, South Nation Nationalities and Peoples Region (SNNPR), and Sidama Regional States. The sample collection sites included Shashemene Referral Hospital, Adama General Hospital and Medical College and Tulu Bolo General Hospital from Oromia Regional State, Arba Minch General Hospital, Wolita Sodo Teaching and Referral Hospital, Nigist Eleni Memorial Hospital, Dilla University Referral Hospital, and Halaba Kulito Primary Hospital from SNNPR, and Yirgalem General Hospital from Sidama Regional state. The specific study hospitals and selection of study participants in the different geographic regions were based on PTB disease prevalence, *M. tuberculosis* lineage distribution, and logistic issues. These hospitals serve not only city residents, but also those living in nearby districts, and patients from a wide catchment area seek medical treatment at these hospitals.

### Study population

The study's source population comprised all adult PTB suspects who had visited selected hospitals in the Central and Southern regions of Ethiopia during the study period. This study included all recently registered smear microscopy and/or Xpert MTB/RIF positive PTB patients, who were 18 years of age or older, provided written informed consent, and met the inclusion criteria. To ensure the accuracy and reliability of the data, patients with serious illnesses who were unable to provide sputum samples were excluded from this study.

## Sputum sample collection

The selected health facilities contacted all individuals who tested positive for PTB, confirmed by smear microscopy and/or Xpert MTB/RIF, before initiating anti-TB treatment. After informing confirmed PTB patients of the protocol, competent medical staff obtained their informed consent at the respective medical facility. The study participants' clinical and socio-demographic data were collected using structured questionnaires. The laboratory staff instructed the patients on how to collect productive sputum samples, which included thoroughly rinsing their mouths and spitting the sputum into a wide-mouthed, leak-proof, and screw-capped container in the sputum collection stand outside the room. The patients carefully handed over the collected sputum samples to the laboratory professional, who confirmed the quality and quantity of the sample before storing it in a 2–8˚C refrigerator for a maximum of five days. During the study period, we collected 350 sputum samples from patients diagnosed with smear microscopy and/or Xpert MTB/RIF positive PTB patients who participated in the study. Finally, the samples were sent to the AHRI TB Laboratory via a cold chain system for culture and DST analysis. AHRI TB Laboratory is one of the TB-culture and DST laboratories in Ethiopia actively involved in the National External Quality Assurance System.

## *M.tuberculosis* growth and identification

Egg-based LJ-pyruvate and LJ-glycerol media were prepared aliquoted and stored in the refrigerator at 2–8˚C for a maximum of two months. The sputum samples were decontaminated by the NALC-NaOH method and centrifuged at 3000 rpm for 15 min. The supernatant was discarded and the deposit was resuspended in 1.5 ml phosphate saline buffer solution. The sediment was inoculated onto conventional LJ media slants supplemented with 0.4% sodium pyruvate and 0.3% glycerol to enhance the growth of the different *M.tuberculosis* complex species and incubated at 37ºC for at least 8 weeks, with weekly observation for the presence of mycobacterial colonies [19]. Microscopic examination of the colonies was performed using Ziehl- Neelsen stain to select AFB-positive isolates. Loop full colonies were harvested into 2 ml cryovials containing 500μl sterile nuclease-free water for heat inactivation. The remaining AFB-positive colonies were collected and frozen in glycerol stocks (freezing media) in duplicate. One of the samples was used for DST, while the other was stored at -80˚C in the central AHRI facility as a backup.

## Molecular typing

The heat-inactivated isolates were studied using PCR-based deletion typing for the presence or absence of RD9 to differentiate *M. tuberculosis* from other mycobacterial species. The sequences of the primers used for RD9 deletion typing were RD9 FF, 5'-GTG-TAG-GTC-AGC-CCC-ATC-C-3', RD9 Int, 5'-CAA-TGT-TTG-TTG-CGC-TGC-3', and, RD9 FR, 5'-GCT-ACC-CTC-GAC-CAA-GTG-TT-3' [20]. PCR amplification of the mixtures was done using a Thermal Cycler PCR machine (Biometra T3000, Thermocycler). The reaction mixture was prepared and amplified using the following program: 10 min at 95˚C for enzyme activation, 1 min at 95˚C for denaturation, 0.5 min at 61˚C for annealing, 2 min at 72˚C for an extension, involving a total of 35 cycles, and a final extension at 72˚C for 10 min. The product was electrophoresed using the Agarose Gel Electrophoresis System in 1.5% agarose gel in 1× Trisacetate- ethylene diamine tetraacetic acid running buffer. Ethidium bromide at a ratio of 1:10, 100 base pair (bp) DNA ladder, and orange 6× loading dye were used in gel electrophoresis and the gel was visualized. The well-characterized laboratory strains *M. tuberculosis* H37Rv and *M.bovis* BCG were used as positive controls, and molecular grade water was used as a negative control.

## Phenotypic drug susceptibility testing

The isolates' phenotypic drug susceptibility patterns were determined with the BACTEC MGIT™ 960 instrument using the SIRE kit. In brief, frozen isolates were thawed and subcultured in MGIT tubes. When the MGIT machine produced a positive signal, it was declared day 0 and DST began the next day. For day 1 and 2 signals, no dilution was performed; however, if the signal was visible on days 3–5, a 1:5 dilution with sterile saline was performed before DST inoculation. If growth was observed after day 5, the samples were vortexed, diluted with sterile saline in a 1:100 ratio, and 0.5 ml was inoculated into the MGIT tube. The experiment was conducted following the established standard operating procedures [21]. DST for streptomycin (STM), INH, RIF, and ethambutol (EMB), resistance was performed according to the WHO technical manual for DST with the following drug concentrations: STM 1.0 μg/ml, INH 0.1 μg/ml, RIF 1.0 μg/ml, and EMB 5.0 μg/ml [22]. The instrument flagged the DST set complete when the growth control reached a growth unit (GU) value of 400. When the GU reaches 400 or higher, and the drug-containing tube reading falls below 100, the test result was reported as "susceptible". On the other hand, if the GU value reaches 400 and the drug-containing tube reads more than 100, the test result was reported as "resistant". If the GU value of the control reaches 400 in less than 4 days and does not reach 400 in 21 days, the result is invalid, and the machine returns an error message X400 and X200 respectively [23]. Quality control was maintained by testing each batch of MGIT medium and SIRE Kit with the pan-susceptible laboratory strain *M. tuberculosis* H37Rv.

## Genotypic drug susceptibility testing

The GenoType® MTBDR*plus* assay is a DNA-STRIP-based molecular genetic assay used to identify RIF- and INH-resistant *M. tuberculosis* isolates. This assay detects the absence and/or presence of wild-type (WT) and/or mutant (MUT) DNA sequences within a specific region of three genes; the promoter region of the *inhA* gene (coding for the NADH enoyl ACP reductase), the *katG* gene (coding for the catalase-peroxidase), and the *rpoB* gene (coding for the β-subunit of the RNA polymerase) enabling the detection of RIF resistance. The test was conducted following the manufacturer's instructions (Hain Life sciences, Nehren, Germany) [24]. Similarly, for MDR-TB isolates, the GenoType® MTBDR*sl*, a DNA-STRIP-based molecular genetic assay, was used to detect resistance to FQs, second-line injectables, and EMB drugs by targeting the *gyrase A*, *rrs*, and *emb* genes respectively [25]. As positive and negative controls, DNA from the standard laboratory reference pan-susceptible *M.tuberculosis* H37Rv strain and molecular-grade water were used.

## Data management and analysis procedure

Demographic and clinical data were recorded using a paper-based case record form (CRF) system at each field site. Subsequently, the CRF was transported to AHRI, where two data encoders independently entered the CRF into the Red Cap database using the double-entry method. Data was then checked for consistency and accuracy. Before exporting the data to a CSV file, all identifiers capable of revealing individual participants were removed to ensure authors had no access to personally identifiable information during the research process. The anonymized CSV data was subsequently imported into IBM SPSS Statistics software (Version 25.0) for further statistical analysis. Descriptive statistics were employed to determine the frequency and percentage of the variables. To assess the agreement between BACTEC MGIT™ 960 DST and GenoType® MTBDR*plus* assay, the kappa value and a 95% confidence interval for the kappa statistic were calculated using MedCal software (Version 20.216).

## Ethical considerations

The AHRI/ALERT Ethics Review Committee (AAERC) reviewed and approved the project (Protocol No. PO/40/20). The study was also reviewed and approved by the Arba Minch University (AMU) Ethical Review Board (Reference No. IRB/1053/21). Before enrolment, each study participant was given information on a standardized information sheet, and the study's objective, risks, and benefits were described to each study participant and, questions were answered. Those who agreed to take part signed an informed consent form and were enrolled in the study. Based on the results, both groups with drug sensitivity and drug resistance were treated at a reputable medical facility.

## Results

### Sociodemographic features of study participants

In this study, sputum samples were collected from 350 newly diagnosed PTB patients, 90% (315/350) were culture-positive. The RD9 deletion typing was performed on all culture-positive isolates, and the results showed that each isolate had an intact RD9 locus and was subsequently identified as *M. tuberculosis* based on previously mentioned bands of varying sizes [20]. Among these isolates, 84.4% (266/315) were successfully sub-cultured and phenotypic DST was performed on the BACTEC MGIT™ 960 instrument using the SIRE kit. These isolates were then heat-killed and their genomes were extracted using the GenoLyse® kit for the Geno-Type® MTBDR*plus* assay to identify INH and RIF-resistant conferring gene mutations. Out of the sub-cultured isolates in the GenoType® MTBDR*plus* assay, 1.9% (5/266) produced invalid results, 4.1% (11/266) were excluded due to missing data during the combined cleaning stage, and 94% (250/266) were included in the paired analysis. Among the study participants (S1 Table), 66.8% of newly diagnosed PTB cases were between the ages of 18 and 34 years. The mean age of the study participants was 31 years (+/- 13SD), whereas the median age was 25 years. Males made up 63.6% of the total participants. The majority of study participants had lower educational attainment, with 41.6% reporting their educational status as grade 1–8 and 29.6% reporting illiteracy. Farmers comprised 26.8%, while students comprised 22.4% of the study participants.

### Phenotypic drug susceptibility patterns of the tested isolates

The study revealed that out of those isolates tested for first-line anti-TB drug susceptibility, 14.4% (36 /250) showed resistance to at least one of the drugs. The remaining 85.6% of the isolates were susceptible to all tested first-line anti-TB drugs, as demonstrated in Table 1. The study examined drug susceptibility patterns of *M. tuberculosis* isolates among newly diagnosed PTB patients and showed that INH had the highest (4.4%) prevalence of mono-resistant isolates, followed by EMB, with 3.2% of isolates showing resistance. In this study, STM mono resistance was found in 0.8% (2/250) of the tested isolates, but no RIF mono resistance was detected. Only 1.6% (4/250) of newly diagnosed pulmonary TB patients had combined drug resistance to INH+STM, 1.2% (3/250) to INH+ETB, and 0.4% (1/250) to STM+EMB. Although no drug-resistant strain was identified in this study for the combination of INH+RIF alone, triple resistance to first-line anti-TB drugs was observed in 2.4% (6/250) of the tested isolates. One isolate (0.4% or 1/250) was resistant to all four anti-TB drugs tested and classified as MDR-TB.

### Mutation patterns of drug-resistant *M. tuberculosis* isolates

The GenoType® MTBDR*plus* assay was utilized to assess the genotypic drug susceptibility patterns of 250 *M. tuberculosis* isolates and detect mutations linked to INH and RIF-resistance

**Table 1. Phenotypic drug susceptibility patterns of *M. tuberculosis* isolates.**

| Tested Drugs | Frequency N (%) |
|---|---|
| **Any resistance to one drug** | |
| Any STM | 14(5.6) |
| Any INH | 25(10) |
| Any RIF | 1(0.4) |
| Any EMB | 19(7.6) |
| **Resistance to only one drug** | |
| STM only | 2(0.8) |
| INH only | 11(4.4) |
| RIF only | 0(0) |
| EMB only | 8(3.2) |
| **Resistance to only two drugs** | |
| INH+STM only | 4(1.6) |
| INH+EMB only | 3(1.2) |
| STM+EMB only | 1(0.4) |
| **Resistant to only three drugs** | |
| STM+INH+EMB | 6(2.4) |
| **Resistant to Four drugs** | |
| STM+INH+RIF+EMB | 1(0.4) |

STM = Streptomycin, INH = Isoniazid, RIF = Rifampicin, EMB = Ethambutol

conferring gene mutations. As indicated in Table 2, 96% of the tested isolates were completely susceptible to both INH and RIF, 3.6% were only resistant to INH, and 0.4% (1/250) was MDR due to resistance to both RIF and INH. Only gene mutations in the *rpoB* and *katG* gene loci were found in the study, signifying RIF and high-level INH resistance. The wild-type (WT) probe in the *rpoB* gene was missing in a single RIF-resistant *M. tuberculosis* isolate and was replaced by the MUT3 probe, which contains a single nucleotide substitution from serine (S) to leucine (L) at position 531. This substitution has been linked to RIF resistance in *M. tuberculosis*, as it results in an altered conformation of the enzyme that is no longer capable of binding to RIF. It is well documented that, INH-resistance has been associated with mutations in two genes: *katG* and *inhA* promoter region, yet only resistance to *katG* gene loci was discovered in this study. The lack of the *katG* WT probe, combined with the hybridization of the *katG* MUT1 probe (Ser315Thr1 substitution), resulted in ten INH-resistant strains demonstrating high-level INH resistance. One isolate displayed positive hybridization of both WT and corresponding MUT1 (Ser315Thr1 substitution) probes of the *katG* gene, signifying a

**Table 2. Resistance pattern, mutation, and amino acid change of drug-resistant *M. tuberculosis* isolates.**

| Tested drugs | Target genes | Amino acid change | Frequency N (%) | Resistance Pattern |
|---|---|---|---|---|
| INH + RIF Susceptible | *rpoB*, *katG*, and *inhA* | NA | 240 (96) | Pan-susceptible |
| INH Resistance | *katG* | S315T1 | 8(3.2) | Monoresistance |
| | | WT+S315T1 | 1(0.4) | Heteroresistance |
| INH + RIF Resistance | *rpoB* | S531L | 1(0.4) | MDR |
| | *katG* | S315T1 | | |

INH = Isoniazid, MDR = Multidrug resistance, MUT = Mutant Type, RIF = Rifampicin, WT = Wild Type

heteroresistance pattern and being classified as a 'rare' mutation. The multidrug-resistant isolate was determined to be caused by a Ser315Thr1 substitution in the *katG* locus, which is a mutation that is widely found all over the world and the primary contributor to the results in this study. The MDR and INH-resistant isolates were then genotypically tested for FQs and SLIDs using the GenoType® MTBDR*sl* assay. There were no mutant genes found, indicating that the MDR and INH-resistant isolates were susceptible to the tested second-line anti-TB drugs.

### Examining the correlation between phenotypic and genotypic drug susceptibility testing

All 250 isolates in this study were subjected to both phenotypic and genotypic DST, allowing us to study paired outcomes. Table 3 summarizes the resolution of phenotypic and genotypic drug resistance discordant results by comparing DST data obtained using two methods. The inter-rater agreement between the two methods, BACTEC MGIT™ 960 and Genotype® MTBDR*plus* assay, for susceptibility to RIF, was 100%, with a kappa value of 1.0 (95%CI 1.0–1.0) while for susceptibility to INH was 94% with a kappa value of 0.56 (95%CI: 0.36–0.76). When compared to the BACTEC MGIT™ DST result, 90% (225/250) of the GenoType® MTBDR*plus* assay results were Susceptible Susceptible (SS) and 4% (10/250) were Resistant Resistant (RR) and were considered concordant pairs. Nevertheless, in this study, 5.6% (14/250) of the pairs displayed Resistant Susceptible (RS) and were considered discordant, and none of the pairs displayed Susceptible Resistant (SR). Using the RS and SR pairs, the overall discordance was 5.6% (14/250) for the INH alone and 0% for RIF alone and MDR-TB detection. When compared to the BACTEC MGIT™ 960 testing method, the GenoType® MTBDR*plus* assay demonstrated high sensitivity and specificity for testing RIF resistance, with a positive predictive value (PPV) and negative predictive value (NPV) of 100%. This means that the test correctly identified 100% of the true positives and 100% of the true negatives. In comparison to the BACTEC MGIT™ DST, GenoType® MTBDR*plus* assay had a low sensitivity of 42% for testing for INH-resistance, but a 100% specificity, with a positive predictive value (PPV) of 100% and a negative predictive value (NPV) of 94.2%. This means that the test correctly identified 94.2% of the true negatives, but only correctly identified 42% of the true positives.

### Discussion

This study explored the phenotypic and genotypic drug susceptibility patterns of *M. tuberculosis* isolates from newly diagnosed PTB patients in Central and Southern Ethiopia. The research

**Table 3. Comparison between phenotypic and genotypic drug susceptibility testing.**

|  | Drugs |  | BACTEC MGIT™ 960 DST (SIRE) | | Total | Diagnostic performance of Genotypic DST | | | | Level of Agreement Kappa (95%CI) |
|---|---|---|---|---|---|---|---|---|---|---|
|  |  |  | Susceptible | Resistant |  | Sensitivity | Specificity | NPV | PPV |  |
| GenoType® MTBDR*plus* assay | RIF | Susceptible | 249 | 0 | 250 | 100% | 100% | 100% | 100% | 1.0 (1.0–1.0) |
|  |  | Resistant | 0 | 1 |  |  |  |  |  |  |
|  | INH | Susceptible | 226 | 14 | 250 | 42% | 100% | 94.2% | 100% | 0.56 (0.36–0.76) |
|  |  | Resistant | 0 | 10 |  |  |  |  |  |  |
|  | MDR | Susceptible | 249 | 0 | 250 | 100% | 100% | 100% | 100% | 1.0 (1.0–1.0) |
|  |  | Resistant | 0 | 1 |  |  |  |  |  |  |

INH = Isoniazid, RIF = Rifampicin, NPV = Negative Predictive Value, PPV = Positive Predictive Value, CI = Confidence Interval

aimed to assess the drug resistance level in the study area and evaluate the discordance between phenotypic and genotypic DST results while exploring the relationship between mutation and drug resistance in *M. tuberculosis* isolates. The study highlights the challenges in accurately identifying drug-resistant strains and identifies the limitations of existing molecular tools, emphasizing the need for more comprehensive approaches to identify drug-resistance-conferring gene mutations in diverse geographical areas.

Surveillance programs play a critical role in monitoring and controlling the spread of TB and drug resistance tuberculosis (DR-TB) strains. To forecast the magnitude in the country, a comprehensive national TB surveillance program is required. It is understood that such a program would require significant investment and should be supplemented by sub-national studies. The findings of studies conducted in various parts of the country can reveal the scope of the problem. In one study, 10% of the tested isolates in the Somali region of Southern Ethiopia were resistant to at least one of the tested anti-TB drugs [26]. In our study in Central and Southern Ethiopia, 14.4% of the tested *M. tuberculosis* isolates showed resistance to at least one anti-TB drug. In contrast, the resistance rate was even higher in the Arsi Zone of Southeastern Ethiopia, with 17.2% of the isolates demonstrating resistance to at least one anti-TB drug [27]. The differences observed between our study and the Somalia region may be due to previous exposure to anti-TB drugs as a community, where the community in the Somalia region has less access to health facilities, implying less access to anti-TB treatment, which minimizes drug-induced resistance, whereas our study community is agrarian, with better access to health facilities. The difference in the Arsi Zone could be due to the area being the first in Ethiopia to pilot a Directly Observed Treatment Short course (DOTs) strategy in 1992 [28] thus maybe having a longer exposure history to anti-TB drugs.

The increasing prevalence of INH mono-resistance poses a potential threat of developing MDR-TB, thereby raising significant concerns for TB control [29]. In our study, we observed a prevalence of INH mono-resistance among 4.4% of the strains. This finding is consistent with a study conducted in the Somali region of Eastern Ethiopia, where the rate of INH mono-resistance was found to be 4.1% [26]. However, similar studies in Ethiopia have reported varying levels of INH mono-resistance, with rates ranging from a low of 1.4% [30] in North East Ethiopia to a high of 9.5% [31] in North West Ethiopia. These disparities in INH mono-resistance levels highlight the need for additional research to determine the underlying factors contributing to the variability. The extensive use of INH as both a TB treatment over the past seven decades and more recently as a prophylaxis drug for HIV patients and bacteriologically confirmed TB contacts may be contributing to this trend [29, 32]. Given these disparities in INH mono-resistance levels and the widespread use of INH, it is essential to have reliable diagnostic tools that can detect INH-resistant strains. Furthermore, this finding is significant as it suggests that INH mono-resistance is a growing problem in the country, and may be caused by the overuse of INH, or a lack of adherence to the guidelines for the proper use of the drug. Additionally, more research needs to be done to determine the exact magnitude of INH-resistant strains in the country.

In our study, we also tested isolates for RIF and INH resistance, which indicates the presence of MDR-TB among circulating *M. tuberculosis* isolates. In this study, 0.4% of the tested isolates were resistant to both INH and RIF, classifying them as MDR-TB strains. This MDR-TB prevalence among newly diagnosed PTB patients appears to be very low when compared to neighboring countries and the global estimates [33–36]. This could be due to a relatively stronger health system in Ethiopia that links Xpert MTB/RIF sites with the TB microscopic sites for sample referral and enables early detection and treatment of RR-TB cases. Unfortunately, the ongoing conflicts in various parts of the country have harmed the health system. Additionally, studies have shown that the COVID-19 pandemic has led to a decrease in health-seeking

behavior among TB patients [37, 38], which could both potentially obscure the true prevalence of MDR-TB in the community. Thus, immediate intervention is required to restore damaged health facilities and improve active case finding to address the missed cases.

The specific gene loci where mutations were found were then examined to determine the mechanisms of DR-TB in circulating *M. tuberculosis* isolates. The *rpoB*, *katG*, and *inhA* gene loci were investigated because they play important roles in the development of resistance to RIF and INH, two commonly used first-line anti-TB drugs. In this study, the absence of *rpoB* WT8 and hybridization of the MUT3 probe (S531L substitution) were discovered in one of the tested isolates, as in previous similar studies [39]. This meant that the tested isolate carried a specific mutation in the *rpoB* gene, which is associated with RIF resistance. Furthermore, the absence of the *katG* WT probe, combined with hybridization of the *katG* MUT1 probe (Ser315Thr1 substitution), was found in ten INH-resistant strains demonstrating high-level INH resistance. Similarly, *Alelign et al.* from the South Gondar Zone in northwest Ethiopia discovered INH-resistance conferring gene mutations in the *katG* gene loci [40] alone. Studies indicated that there are significant variations in the occurrence of *katG* and *inhA* mutations among *M. tuberculosis* strains globally, with *katG* mutations consistently reported to be more prevalent than *inhA* mutations [41]. The absence of mutations in the structural region of the *inhA* gene implies that a separate and distinct mutation in its promoter region may regulate INH resistance in *M. tuberculosis* isolates [42]. A systematic review and meta-analysis of selected studies from various parts of the country revealed that among the circulating *M. tuberculosis* complex strains, mutations in the *katG* gene S315T substitution for INH resistance and *rpoB* gene S531L substitution for RIF resistance-conferring gene mutations were found in high magnitude among the other tested regions, indicating that this type of mutation is very common in Ethiopia [43, 44]. This discovery shows that the widely used GenoType® MTBDR*plus* assay in the country is more capable of detecting such changes in probes. Given that the GenoType® MTBDR*plus* assay is extensively utilized in referral and research laboratories in Ethiopia, knowing this crucial information can aid in treatment decisions and the selection of the most appropriate drugs.

During our study, in one of the tested isolates we identified an INH-heteroresistance pattern, wherein both the wild-type (WT) and corresponding MUT1 probes of the *katG* gene tested positive, indicating a rare occurrence. According to an Ethiopian population-based drug resistance survey, INH-heteroresistant strains were present in a significant proportion, with 70% of such strains identified in newly diagnosed TB patients [45]. Heteroresistance can occur in two ways: through mixed infections in which both resistant and susceptible strains infect a person at the same time, or through the development of resistance through genetic mutation in a single clone of a previously susceptible strain when subjected to antibiotic pressure. This pattern is regarded as an initial step to change from susceptible to mono-resistant and/or MDR [46], posing a potential threat to the successful treatment of INH-resistant patients and potentially increasing the risk of anti-TB drug resistance in the study area. It is critical to monitor and detect such rare mutations promptly, and more research is needed to understand the mechanisms underlying heteroresistance patterns and their implications for TB control strategies.

The major findings of this study showed the presence of INH discordant isolates, where the strain was phenotypically INH-resistant but had the wild-type gene. This phenomenon was observed in 5.6% of the isolates, which indicated that the two methods of identifying INH-resistant *M. tuberculosis* isolate disagreed with each other. This means that the GenoType® MTBDR*plus* assay has a sensitivity of 42% and a specificity of 100% in detecting INH-resistant *M. tuberculosis* isolates, with a kappa value of 0.56 (95%CI: 0.36–0.76) when compared to the BACTEC MGIT™ 960 DST. Multiple reports from various parts of the country and around the world revealed varying results regarding the agreement between the two methods for diagnosing DR-TB [17, 18, 31, 47, 48]. The availability of WHO-approved rapid molecular diagnostic

tools, such as the GenoType® MTBDR*plus* assay [49], even in limited settings like public referral laboratories and research institutions has revolutionized the diagnosis of DR-TB in Ethiopia. These tools have helped generate data and provide valuable insights for public health efforts in the country. However, the specific gene mutations that confer drug resistance can differ depending on the geographic location, so the use of comprehensive probes is necessary to cover all potential drug-resistant conferring gene mutations. According to *Farhat et al.*, diagnostic technology that is equipped with more drug-resistant conferring gene mutations can provide a comprehensive DR-TB diagnosis [50]. In this regard, conducting regular monitoring of drug resistance-conferring gene mutations, and testing new probes and algorithms is needed as drug resistance is not a static phenomenon, but can increase or spread over time.

This study had limitations in tracking treatment outcomes for patients diagnosed as phenotypically resistant but carrying wild-type genes. The collection of samples in hospitals necessitated patient referral to the nearest health facility for directly observed therapy, creating challenges for follow-up. This study also acknowledged the limitations of phenotypic methods for EMB testing and the potential risk of false resistance interpretation in MGIT DST, as outlined in the WHO Technical Manual for DST. Furthermore, resource constraints prevented us from conducting repeated INH MGIT on the strains with discordant results to confirm INH resistance. These constraints also prevented us from conducting MGIT second-line DST, thereby limiting the ability to compare the phenotypic and genotypic DST patterns of the isolates for second-line drugs. Lastly, the limited number of study participants may restrict the generalizability of the findings, potentially underestimating the complexity and extent of the problem in the broader population.

## Conclusions

This study emphasizes the importance of being vigilant in monitoring drug resistance to detect emerging DR-TB strains, assess control measures, and track progress toward global targets for TB control and elimination. INH mono-resistance and INH discordant isolates were particularly prevalent, highlighting the need for robust testing methods to accurately detect drug resistance. To this end, it is important to use a combination of genotypic and phenotypic methods to predict novel drug resistance targets and to test new probes and algorithms. Furthermore, reliable methods, such as whole genome sequencing, should be used to identify representative INH resistance-conferring point mutations in Ethiopia that can be included in the WHO-approved rapid molecular tools. This will help to improve the accuracy of molecular tools used to detect drug resistance in *M. tuberculosis* isolates and ensure that the most effective treatments for TB infections are prescribed.

## Supporting information

**S1 Table. Sociodemographic feature of the study participants in central and southern Ethiopia.**
(DOCX)

**S1 Raw data.**
(CSV)

## Acknowledgments

The authors would like to express their heartfelt gratitude to the Regional Health Bureaus of Oromia and the South Nation Nationalities and Peoples for their unwavering support in facilitating the conduct of this study in their respective hospitals. The authors also express

heartfelt gratitude to the Zonal Health Administrators, Chief Executive Officers, Laboratory, and TB Clinic staff at all participating hospitals for their invaluable assistance in ensuring the project's successful implementation. Furthermore, the authors wish to express gratitude to the study participants, who selflessly contributed their time, effort, and samples to the project. Their willingness to collaborate and share valuable information was critical to the project's success.

## Author Contributions

**Conceptualization:** Melaku Tilahun, Abraham Aseffa.

**Data curation:** Melaku Tilahun, Tewdros Tariku Gebresilase, Bethlehem Adnew, Maeruf Fetu Nassir.

**Formal analysis:** Melaku Tilahun, Tewdros Tariku Gebresilase.

**Funding acquisition:** Yonas Kassahun, Abraham Aseffa, Kidist Bobosha.

**Investigation:** Melaku Tilahun, Tesfaye Gebreyohannes, Abraham Tekola, Mekdes Alemu, Sebsib Neway.

**Methodology:** Melaku Tilahun, Yonas Kassahun, Abraham Aseffa.

**Project administration:** Tewdros Tariku Gebresilase.

**Supervision:** Teklu Wegayehu, Biniam Wondale, Kidist Bobosha.

**Validation:** Melaku Tilahun.

**Visualization:** Melaku Tilahun.

**Writing – original draft:** Melaku Tilahun.

**Writing – review & editing:** Teklu Wegayehu, Biniam Wondale, Tewdros Tariku Gebresilase, Tesfaye Gebreyohannes, Abraham Tekola, Mekdes Alemu, Sebsib Neway, Bethlehem Adnew, Maeruf Fetu Nassir, Yonas Kassahun, Abraham Aseffa, Kidist Bobosha.

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
