## [Decision Letter · Decision Letter 0]

18 Jun 2023

PONE-D-23-09720Phenotypic and genotypic drug susceptibility patterns of Mycobacterium tuberculosis isolates from pulmonary tuberculosis patients in Central and Southern EthiopiaPLOS ONE

Dear Dr. Tilahun,

Thank you for submitting your manuscript to PLOS ONE. After careful consideration, we feel that it has merit but does not fully meet PLOS ONE’s publication criteria as it currently stands. Therefore, we invite you to submit a revised version of the manuscript that addresses the points raised during the review process.

We look forward to receiving your revised manuscript.

Kind regards,

Laura Ellen Via, Ph.D.

Academic Editor

PLOS ONE

Journal Requirements:

"This work is supported by Human Heredity and Health in Africa (H3Africa) [Grant number: H3A-18-003] . H3Africa is managed by the Science for Africa Foundation (SFA Foundation) in partnership with Wellcome, NIH, and AfSHG. Part of the work also supported by AHRI Core budget. AHRI receives core support from Sida, NORAD and the Government of Ethiopia."

Additional Editor Comments:

I have also provided independent comments due to the difficulty in finding a second reviewer. I would like to see answers to all the provided comments with special attention to either performance of discordance testing if the strains are still available or consideration of the limitation of the MGIT testing performance if no repeat testing is possible for the strains that had resistant results and a select set of the S strains. 

Comments 

The authors have used convenience sampling to collect a set of septum samples from newly diagnosed MTB infected subjects. The manuscript is well prepared and written but I have a few comments with the last being the most substantive. From the supplemental data set, there were 36 DR strains, but only 10 with a genotypic explanation. I suggest providing the data set in a .csv file in future so that it is ease to review. Further there were 25 strains that tested as INH-R in MGIT 0.1 ug/mL while only of those 9 had KatG mutations and 1 mixed genotype KatG/WT was detected. No inhA mutations were detected, leaving 14 strains with discordant results. AhpC-oxyR intergenic region was not tested. The streptomycin resistance was also not verified molecularly although since the drug is little used and has minimal overlap with other injectables, it is likely not significant to the subjects treatment, but I would have preferred to see these strains molecularly confirmed.

One MDR strain was identified by both genotypic and phenotypic testing. Generally, the definition of MDR TB requires both INH and RIF resistance, so only 1/250 strains tested were MDR. The other 5 strains might be better called something else, maybe polyresisent or another term in common use.

The WHO Technical manual for drug susceptibility testing of medicines for treatment of tuberculosis (2018) now recommends a 0.1 ug/mL for INH. Often the 0.4 mg/mL INH concentration has been used and is useful in that it identified INH highly resistant strains. We have found the 0.4 ug/mL concentration correlates better with the occurrence of the katG mutation. This same document indicates that all phenotypic methods for EMB resistance produce inconsistent results and it is not recommended. I would suggest that without genotypic evidence of EMB resistance, this data not be emphasized. You might list this poor test performance in the limitations.

No discussion is included of the proficiency of the MGIT testing laboratory. Does it participate in a TB DST qualification program on a regular basis?

There are numerous reports of mixed strains that have a WT and MUT phenotype. Not sure how often it is seen in Africa but its common in Asia. Not sure I would use the word surprising in the results, especially since you discuss others seeing mixed signals in the discussion.

Lines 290 to 292 – it is not clear which strains were tested with the MTB-DRplus assay. Was it all strains, if 7 strains or 1, perhaps indicate in table with a subscript. Part of the confusion is the definition of MDR. The authors are also confused since they use both a singular and plural form of isolate(s) in these lines.

I am not convinced that table 3 is necessary given the amount of data in it and it well described in the text, but if it is kept, I suggest using the mutation code for amino acid change (column 3) and not the names of the probes from the testing kits. WT/ can be added for the mixed strain.

In line 318-319, the authors talk about the performance of MTBDRplus assay but do not indicate that any repeat discordance MGIT testing was done. Slight differences in drug inoculation and bacterial inoculums produce results indicating resistance that can be in error. Repeated MIGT INH testing as a discordance evaluation is advised for the purposes of a manuscript like this, if you want to discuss test performance.

Reviewers' comments:

Reviewer's Responses to Questions

**Comments to the Author**

1. Is the manuscript technically sound, and do the data support the conclusions?

Reviewer #1: Partly

2. Has the statistical analysis been performed appropriately and rigorously? 

Reviewer #1: Yes

3. Have the authors made all data underlying the findings in their manuscript fully available?

Reviewer #1: No

4. Is the manuscript presented in an intelligible fashion and written in standard English?

Reviewer #1: Yes

5. Review Comments to the Author

Reviewer #1: The authors provide an overview of pDST and gDST results from a cross-sectional study of TB isolates from Central/South Ethiopia.

1. The major finding of the study was noted that there was discordance between pDST and gDST for INH (14 isolates), and the authors concluded that INH-R TB was found in Ethiopia that was not detected by LPA. However this conclusion was not supported by any discordance testing therefore it is unknown whether these difference were due to improper MGIT DST performance or actual mutations not found on the LPA. Therefore this is a major flaw of the study, and discordancy testing (if not possible to be performed) should be addressed.

2. Why was MGIT DST performed only for the low concentration (0.1 ug/mL) and not the 0.4 ug/mL concentration? Considering the 100% of INH mutations to be katG, it would have been useful to test this higher concentration.

3. There were no inhA mutations detected-this is a bit unusual considering low-level INH resistance is commonly found. The authors should include this in the discussion.

4. Second-line LPA testing was only performed on the MDR isolate (N=1)-however to fully understand the resistance profiles of the isolates tested it would have been useful to employ this testing on all isolates (or at least those with any resistance) to look at FQ monoresistance and correlate with MGIT 2nd line DST.

5. Supplementary material was not able to be reviewed as the file would not download (It was in a .SAV format)

6. The demographic information, while useful to review was not the main focus of the study and can be put as a supplemental Table.

7. Kindly check spelling of "heteroresistance" and "MGIT" throughout manuscript.

6. PLOS authors have the option to publish the peer review history of their article (what does this mean?). If published, this will include your full peer review and any attached files.

Reviewer #1: No

---

## [Author Response · Author response to Decision Letter 0]

31 Jul 2023

Responses to Reviewers

I. Comments

Comment: The authors have used convenience sampling to collect a set of septum samples from newly diagnosed MTB-infected subjects. The manuscript is well prepared and written but I have a few comments with the last being the most substantive. From the supplemental data set, there were 36 DR strains, but only 10 with a genotypic explanation. I suggest providing the data set in a .csv file in future so that it is easy to review. Further there were 25 strains that tested as INH-R in MGIT 0.1 ug/mL while only of those 9 had KatG mutations and 1 mixed genotype KatG/WT was detected. No inhA mutations were detected, leaving 14 strains with discordant results. AhpC-oxyR intergenic region was not tested. The streptomycin resistance was also not verified molecularly although since the drug is little used and has minimal overlap with other injectables, it is likely not significant to the subjects treatment, but I would have preferred to see these strains molecularly confirmed.

Response: Apologies for the inconvenience caused by the inaccessible.SAV format. We have now uploaded the supplementary material in .csv format. Due to budget constraints, we were not able to perform whole genome sequencing which has been reflected as a recommendation of this study . 

Comment: One MDR strain was identified by both genotypic and phenotypic testing. Generally, the definition of MDR TB requires both INH and RIF resistance, so only 1/250 strains tested were MDR. The other 5 strains might be better-called something else, maybe polyresisent or another term in common use.

Response: Comment accepted and corrected accordingly 

Comment: The WHO Technical Manual for drug susceptibility testing of medicines for the treatment of tuberculosis (2018) now recommends 0.1 ug/mL for INH. Often the 0.4 ug/mL INH concentration has been used and is useful in that it identified INH highly resistant strains. We have found the 0.4 ug/mL concentration correlates better with the occurrence of the katG mutation. This same document indicates that all phenotypic methods for EMB resistance produce inconsistent results and it is not recommended. I would suggest that without genotypic evidence of EMB resistance, this data not be emphasized. You might list this poor test performance in the limitations.

Response: Comment accepted and the concerns are included in the limitation section. 

Comment: No discussion is included of the proficiency of the MGIT testing laboratory. Does it participate in a TB DST qualification program on a regular basis?

Response: Comment accepted and corrected accordingly . 

Comment: There are numerous reports of mixed strains that have a WT and MUT phenotype. Not sure how often it is seen in Africa but it is common in Asia. Not sure I would use the word surprising in the results, especially since you discuss others seeing mixed signals in the discussion.

Response: Thank you for your suggestion. Comment accepted and corrected accordingly.

Comment: Lines 290 to 292 – it is not clear which strains were tested with the MTB-DRplus assay. Was it all strains, if 7 strains or 1, perhaps indicate in table with a subscript? Part of the confusion is the definition of MDR. The authors are also confused since they use both a singular and plural form of isolate(s) in these lines.

Response: Thank you for pointing out the typographical error. We have revised the sentence accordingly . 

Comment: I am not convinced that table 3 is necessary given the amount of data in it and it well described in the text, but if it is kept, I suggest using the mutation code for amino acid change (column 3) and not the names of the probes from the testing kits. WT/ can be added for the mixed strain.

Response: Thank you for your suggestion. We agree with your comment and have revised the table accordingly.

Comment: In line 318-319, the authors talk about the performance of MTBDRplus assay but do not indicate that any repeat discordance MGIT testing was done. Slight differences in drug inoculation and bacterial inoculums produce results indicating resistance that can be in error. Repeated MIGT INH testing as a discordance evaluation is advised for the purposes of a manuscript like this, if you want to discuss test performance.

Response: Thank you for your suggestion. The primary objective of this study is to provide an accurate and unbiased report of our findings. We diligently followed the national tuberculosis referral laboratory standard operating procedure (SOP) to determine the phenotypic and genotypic drug susceptibility patterns of M. tuberculosis isolates in study participant samples. As you pointed out, we acknowledge that even slight variations in drug and inoculum concentrations can influence the phenotypic DST results. Furthermore, we suspect that biofilm formation might also contribute to these variations. To address and comprehend these discrepancies, we are actively engaged in a project to identify the underlying causes, and we plan to present our findings in a series of future works. Additionally, we have taken into account the limitations of the MGIT DST as outlined in the WHO DST technical manual (2018). We have made necessary adjustments to the manuscript to appropriately address these concerns and ensure the validity and reliability of our study .

Reviewer Comment

Comment: The major finding of the study was noted that there was discordance between pDST and gDST for INH (14 isolates), and the authors concluded that INH-R TB was found in Ethiopia that was not detected by LPA. However, this conclusion was not supported by any discordance testing therefore it is unknown whether these differences were due to improper MGIT DST performance or actual mutations not found on the LPA. Therefore, this is a major flaw of the study, and discordancy testing (if not possible to be performed) should be addressed.

Response: Thank you for your feedback. We have now updated the conclusion of the manuscript accordingly .

Comment: Why was MGIT DST performed only for the low concentration (0.1 ug/mL) and not the 0.4 ug/mL concentration? Considering the 100% of INH mutations to be katG, it would have been useful to test this higher concentration.

Response: Thank you for your valuable comment. We would like to acknowledge that our study adhered to the national tuberculosis reference laboratory standard operating procedure (SOP) in Ethiopia, which specifically recommended a low concentration (0.1 ug/mL) for MGIT DST. By following this guideline, our study ensured alignment with local practices and maintained consistency with the established protocols in the country.

Comment: There were no inhA mutations detected-this is a bit unusual considering low-level INH resistance is commonly found. The authors should include this in the discussion.

Response: We greatly appreciate your input, as it has helped us improve the manuscript. We have carefully considered your feedback and made the necessary updates 

Comment: Second-line LPA testing was only performed on the MDR isolate (N=1)-however to fully understand the resistance profiles of the isolates tested it would have been useful to employ this testing on all isolates (or at least those with any resistance) to look at FQ monoresistance and correlate with MGIT 2nd line DST.

Response: Thank you for your comment. The genotypic testing for fluoroquinolones (FQ) and SLIDs has been conducted on the INH-resistant isolates. The results have been included in the manuscript . However, due to resource limitations, our laboratory is unable to perform second-line MGIT DST, which is acknowledged as a study limitation.

Comment: Supplementary material was not able to be reviewed as the file would not download (It was in a . SAV format)

Response: Apologies for the inconvenience caused by the inaccessible.SAV format. We have now uploaded the supplementary material in .csv format, which is easily downloadable. Thank you for your understanding and feedback.

Comment: The demographic information, while useful to review was not the main focus of the study and can be put as a supplemental Table.

Response: Thank you for your suggestions. We appreciate your feedback, and as per your recommendation, we have accepted and uploaded Table 1 as a supplementary file. 

Comment: Kindly check the spelling of "heteroresistance" and "MGIT" throughout the manuscript.

Response: Thank you for bringing the typographical error to our attention. We have thoroughly reviewed the manuscript and made the necessary revisions to correct the errors throughout the text .

---

## [Editor Report · Decision Letter 1]

23 Aug 2023

Phenotypic and genotypic drug susceptibility patterns of Mycobacterium tuberculosis isolates from pulmonary tuberculosis patients in Central and Southern Ethiopia

PONE-D-23-09720R1

Dear Dr. Tilahun

We’re pleased to inform you that your manuscript has been judged scientifically suitable for publication once the two comments below are addressed and will be formally accepted for publication once it meets all outstanding technical requirements.

Kind regards,

Laura Ellen Via, Ph.D.

Academic Editor

PLOS ONE

Additional Editor Comments (optional):

I thank you for your responses to the reviewers comments. Other than a few small remaining clarifications, the responses are accepted.

On line 450, I'd suggest removing the words "due to contamination",

On line 451 I'd suggest a sentence more like "resource constraints prevented us from

conducting repeated INH MGIT on the the strains with discordant results to confirm INH resistance. These constraints also prevented us from....."
---

## [Editor Report · Acceptance letter]

29 Aug 2023

PONE-D-23-09720R1 

Phenotypic and genotypic drug susceptibility patterns of *Mycobacterium tuberculosis* isolates from pulmonary tuberculosis patients in Central and Southern Ethiopia 

Dear Dr. Tilahun:

I'm pleased to inform you that your manuscript has been deemed suitable for publication in PLOS ONE. Congratulations! Your manuscript is now with our production department. 

Kind regards, 

on behalf of

Dr. Laura Ellen Via 

Academic Editor

PLOS ONE